# Cancer Stem Cells and Prostate Cancer: A Narrative Review

**DOI:** 10.3390/ijms24097746

**Published:** 2023-04-24

**Authors:** Yazan Al Salhi, Manfredi Bruno Sequi, Fabio Maria Valenzi, Andrea Fuschi, Alessia Martoccia, Paolo Pietro Suraci, Antonio Carbone, Giorgia Tema, Riccardo Lombardo, Antonio Cicione, Antonio Luigi Pastore, Cosimo De Nunzio

**Affiliations:** 1Urology Unit, Department of Medico-Surgical Sciences & Biotechnologies, Faculty of Pharmacy & Medicine, Sapienza University of Rome, 04100 Latina, Italy; 2Urology Unit, Sant’Andrea Hospital, Sapienza University of Rome, 00189 Rome, Italy

**Keywords:** prostate cancer, stem cells, epigenetic, microRNA

## Abstract

Cancer stem cells (CSCs) are a small and elusive subpopulation of self-renewing cancer cells with the remarkable ability to initiate, propagate, and spread malignant disease. In the past years, several authors have focused on the possible role of CSCs in PCa development and progression. PCa CSCs typically originate from a luminal prostate cell. Three main pathways are involved in the CSC development, including the Wnt, Sonic Hedgehog, and Notch signaling pathways. Studies have observed an important role for epithelial mesenchymal transition in this process as well as for some specific miRNA. These studies led to the development of studies targeting these specific pathways to improve the management of PCa development and progression. CSCs in prostate cancer represent an actual and promising field of research.

## 1. Introduction

Prostate cancer (PCa) represents the most common solid tumor among men in Western countries and the fifth cause of cancer mortality worldwide [1]. According to the actual statistics, the worldwide prostate cancer burden is expected to grow to almost 2 million new cases and 700,000 deaths in the next 20 years [2]. Age, race, genetics, family history, and lifestyle (e.g., obesity, smoking, etc.) represent risk factors for higher incidence of PCa [1]. Early detection of PCA is possible through routine checkups, with the determination of prostate specific antigen (PSA), through digital rectal examination, mpMRI, and prostate biopsy [3]. Several risk calculators/apps are available in order to estimate the risk of PCa and significant PCa [4,5].

Several studies have investigated the etiology and pathophysiology of PCa, but the exact mechanisms are still unknown [6]. Important roles are played by the genetic characteristics of the patients as well as its immune system [7,8,9]. The heterogeneous biological nature of the disease brings great challenges during treatment, highlighting the need for better definition of the pathophysiology of PCa [10].

In the past years, several authors have focused on the possible role of stem cells in prostate cancer. Stem cells have the ability to copiously proliferate (self-renewal) starting from a single cell (clonal) and to differentiate into various kinds of cells and tissue (potent) [10]. Self-renewal and potency characteristics are very similar to those of cancer cells [10]. This led to the hypothesis that tumors may arise from the transformation of stem cells and that cancer stem cells (CSC) found inside tumor tissues may be responsible for tumorigenesis [11].

Prostate cancer stem cells (PCSCs) were discovered for the first time by Collins et al. in 2005 in human prostate tumors, and the authors enhanced an increased ability of proliferation in PCSCs, highlighting the idea that they may play an important role in the pathogenesis of prostate cancer. PCSCs have been isolated in metastatic PCa cell lines [11]. Since their discovery, PCSCs have been of great interest in the study of the PC. Today, different and various antigens are known to be responsible for the carcinogenesis of PCa as well as the formation of metastasis and the resistance to pharmacological treatment [12].

The CSC model hypothesis explains tumor heterogeneity, initiating ability, and therapeutic resistance [13]. In addition, a small number of CSCs are needed to recapitulate the tumor and its initial heterogeneity [13]. Cancer cells are organized in hierarchical order and CSCs stem on top of the pyramid, having stem-like characteristics such as self-renewal, pluripotency, and plasticity, that evolve during the lifetime of a tumor [14]. CSCs may be involved in PCa development and may solve the unmet needs in PCa prevention and diagnosis [13]. Their possible role in PCa progression may also improve the current gaps in PCa treatment, especially in the castration-resistant stages of the disease when androgen sensitivity is compromised [15]. Although several treatments have been introduced in this setting, there are still several unmet needs [16,17].

An overview of the most important evidence on this complex topic is essential, especially for clinical urologists. The aim of the present review is to summarize in a narrative fashion the most relevant data on CSCs and PCa.

## 2. Evidence Synthesis

### 2.1. Prostate Cancer Stem Cells Origin and Phenotype

PCa is an heterogeneous disease where different grades may coexist [18,19]. Cell heterogeneity, upon progression and treatment, tends to accentuate [18,19]. The best example of this process is the expression of the androgen receptor (AR), where AR−PCa cells gradually become, in high-grade untreated tumors, the predominant cell population in castration-resistant PCa (CRPC) [18,19].

The first prospective studies aimed to identify PCSCs markers in patient-derived PCa cell lines [11]. These PCa cells expressed surface markers such as CD44+/α2β1hi/CD133+ and the expression of transporter protein ATP-binding cassette subfamily G member 2 (ABCG2) was identified [18].

Since 2005, several markers have been isolated for PCSCs, such as CD44/CD133, aldehyde dehydrogenase (ALDH), CD166, and CD44+/CD24−. PCa cells expressing these markers can self-renew and generate heterogeneous cell subpopulations [20,21]. Furthermore, markers such as CD151, CD160, and podocalyxin TRA-60–1 are associated with PCSCs that give birth to tumors with hierarchical organization cell organization [22]. Expression of SOX2, Notch, and Oct4 are associated with epithelium–mesenchymal transition, which promotes PCa cell migrations [23,24].

PCSCs correlated markers are shown in Table 1.

The origin of PCSCs is debated. In a normal prostate, the stem cell population resides in the proximal duct, originating from the basal cell layer of the epithelium [25]. Moad M. et al. proved in a recent gene-tracing study that luminal cells have a stem cell population, though less potent than the basal stem cell population [26]. PCSCs might derive from the basal cell layer because p63 and other basal markers are expressed in tissue-derived tumor-initiating cells in immunocompromised mice but did not express the AR or other markers of luminal differentiation (PSA and PAP) [27]. PCSCs might originate from normal prostate stem cells after epigenetic modifications created by the tumor microenvironment [28]. *MYC* activation and *PTEN* loss of function added to mutations to DNA repair genes such as *BRCA2* create instability to the genome and cause tumor heterogeneity and progression [29,30]. It is well known that common gene fusion in differentiated PCa is represented by TMPRSS2: ERG. Recent studies by Polson and colleagues also showed that PCSCs had high-frequency of TMPRSS2:ERG [31]. ERG is a transcription factor under the control of TMPRSS2. TMPRSS2 is an androgen-regulated, prostate-specific gene promoter, which is upregulated in CSCs. The result of its activation is differentiation, self-renewal, and maintenance of SCs [32]. Another important marker of CSCs lies in the Homebox gene Nkx 3.1 during prostate regeneration. More specifically, animal studies demonstrate that Nkx 3.1 is required for CSC maintenance and that the deletion of PTEN in these cells results in a rapid transformation in castration-resistant carcinoma [33,34].

### 2.2. PCSC Molecular Pathways and Metabolism

In order to better uncover new and improved therapeutic strategies, molecular pathways and PCSC metabolism have to be analyzed. The Wnt, Sonic Hedgehog, and Notch signaling pathways have been found to be crucial for CSC development [35,36].

#### 2.2.1. Wnt Pathway

The normal Wnt pathway affects cell survival; Wnt ligands bind to Frizzled and low-density lipoprotein receptor-related protein (LRP) 5/6, activating the downstream of the subsequent molecular targets, accumulating β-catenin and mediating its translocation in the cell nucleus [34]. Wnt activates downstream effectors and activates targeted gene expression and cytoskeleton rearrangement, resulting in altered cell survival in noncanonical pathways [34]. In PCA, not only elevated β-catenin expression is found in the cancer cell nucleus [35], but Wnt signal promotes cell self-renewal in several cell models including LNCaP, C42B, and PC3 cells in an AR-independent way, increasing expression of CD133 and CD44 [36]. The downregulated Wnt/β-catenin pathway significantly suppresses stem cell-like properties instead [37]. Furthermore, Wnt3 has been shown to increase the expression of its downstream effectors and CSC markers, including CD133 and CD44, thus playing an important role in CSC development in CRPC (Figure 1).

#### 2.2.2. Sonic Hedgehog Pathway

The Sonic Hedgehog signaling pathway controls cell renewal and survival. The ligands of this family are represented by Sonic, Desert, and Indian. These ligands, by binding to membrane receptors, Patched (Ptch1 and 2) and Smoothered, activate and mediate the nuclear translocation of glioma-associated oncogene homolog (Gli), expressing target genes that regulate cell survival. An abnormal Hedgehog signaling pathway is present in PCa and in CSC proliferation [38,39]. Gli-related genes, including *CDKN2A/p16/INK4A*, *Myc*, and *CDK2*, by promoting androgen-independent tumor cell growth, lead to biochemical and tumor recurrence and lead to tumor recurrence [40]. Furthermore, cell models have shown that FOXA1 and BCL2 have a role both in CRPCa and therapeutic resistance [41]. Hedgehog signaling was also found to be increased. Hedgehog signaling was found with PCSC CD44+, CD24− [21] (Figure 2).

#### 2.2.3. Notch Signaling Pathway

The Notch signaling pathway is mediated by the Notch1-4 receptors and from ligands, such as DLL 1, DLL 3, DLL4, Jagged 1, and Jagged 2. In PCa Notch, the AR pathway and the PI3K/Akt pathway interact. Importantly, Notch interacts with the AR pathway and the phosphoinositide 3-kinase (PI3k)/Akt pathway, which are the two main signaling pathways that regulate development and carcinogenesis [42,43]. Increased levels of Jagged 1-Notch is associated with PCa progression, metastasis, and EMT, and increased Notch3 expression is CRPCA [44]. PC tumor development in the beginning stages is dependent on androgen signaling. Initially, most patients respond to ADT, but almost the totality of them develop CRPC, hence the shift to androgen-independent tumors [45]. It has been found that AR-variants (AR-V) present a truncation in the COOH terminal, thus lacking the ligand-binding domain but maintaining transcriptional capability. These AR-Vs are associated with PCa growth and progression. In particular, AR-V7 has been found in CTCs and in PC resistant to new antiandrogenic drugs such as enzalutamide and abiraterone [46] (Figure 3).

#### 2.2.4. Metabolism of PCSC

The normal luminal cell compartment of the prostate gland produces and secretes a large amount of citrate into the prostatic fluid to nourish and preserve sperm vitality. Normally citric acid represents an important substrate in the tricarboxylic acid cycle, being transported from the mitochondria, where it is produced, to the cell via transmembrane transporters [47]. In luminal cells, mitochondrial aconitase has low activity because of the expression of zinc transporter proteins, which keep elevated zinc levels, impeding citrate oxidation. High production of citrate in the prostate gland is a consequence of the low activity of mitochondrial aconitase (m-ACNT) and subsequent inhibition of citrate oxidation [47].

On the other hand, PCa cells present low levels of zinc in the mitochondria, inactivating the inhibitory effect on m-ACNT, thus enabling the use of citrate as a metabolic substrate [48], oxidizing citrate, and permitting oxidative phosphorylation. Further evidence has shown that the shift from oxidative phosphorylation to glycolysis is caused by mutations in mtDNA of the oncogenic suppressors *PTEN* and *p53* [49]. This shift is needed due to the intrinsic properties of tumor cells, being fast-growing and metabolically active. This phenomenon, also known as the Warburg effect, is especially evident in metastasized CRPCA, where ROS produced and released by PCa cells cause oxidative stress in CAF, consequently shifting to glycolysis and producing high levels of lactate, which fuels ATP production [50] (Figure 4).

It is well known that tumor cells, particularly fast-growing PCa cells, utilize glutamine as one of the main metabolic substrates. Tissues with enhanced expression of MYC oncogene, such as PCa, are particularly dependent on glutamine metabolism to sustain their viability [51]. *MYC* is involved in glucose metabolism, regulating GLUT1, HK2, PFK1, enolase 1, and LDHA [16]. In addition to its role in glucose metabolism, MYC regulates the expression of GLS1, and genes such as SLC1A4 and SLC1A5 [52]. MYC involvement is well established in PCa: for example, the upregulation of MYC mRNA is present in most PCa; furthermore, its overexpression is present in about 30% of CRPC [53].

### 2.3. Epithelial–Mesenchymal Transition and CSC Correlation

Epithelial–mesenchymal transition (EMT) is a cellular process in which normal epithelial cells lose their epithelial features, transitioning in pseudo-mesenchymal cells [54]. EMT normally occurs in the embryonic phase of development, from before implantation to organogenesis. Tumor cells activate these processes once more, leading the cells to acquire motility and invasiveness potential. During prostate cancer progression, the epithelial cells can undergo EMT, characterized by morphological changes in their phenotype from cuboidal to spindle-shaped. Malignant cells lose E-cadherine (E-cad), syndecans, and tight-junction molecules while under the influence of SNAL1, SNAI2/Slug, and TWIST. On the other side, the expression of mesenchymal cell markers Vimentine, N-cadherin, and metalloproteinases are upregulated in malignant cells. Furthermore, the main causal factor for PCa progression is the loss of E-cadherin. This process results in the formation of a cell phenotype with increased migration and invasion potential as well as metastatization [55].

#### 2.3.1. Loss of E-Cadherin Expression

Overall, the EMT process is a complex genetic program, involving numerous interactions among several different EMT-transcription factors (EMT-TFs). Molecules related to these pathways include TGFβ, FGFRs, and PDGF. The primary EMT-TF families that downregulate E-cad expression are *Zeb (Zeb1/Zeb2); Snail (Snail/Slug)*, and *Twist1* [56]. Androgens are also involved in EMT. Mesenchymal phenotypes can be induced since androgens have the ability to suppress E-cadherin in normal prostatic cells. Zhu et al. have proven that elevated levels of DHT can facilitate EMT. DHT alone and in combination with TGFβ induces the expression of Snail leading to the reduction in E-cadherin and β-cathenin levels. In addition, Zhu et al. also evaluated AR levels, finding that AR levels influence EMT. In the presence of a high concentration of AR, DHT exposure did not cause the reduction in epithelial markers, while in cells with low AR concentration levels, the opposite effect was found, not only DHT downregulated E-cadherin, but also increased the cells’ invasion potential. AR loss is normally caused by androgen deprivation. To this date, ADT is the main medical treatment for PCa, but it has been speculated that the treatment might fuel EMT [57] (Figure 5).

#### 2.3.2. Role of TGF-β

In normal circumstances, TGFβ’s role is that of tumor suppressor promoting apoptosis. Cancer cells, either inactivating TGFβ’s receptors or repressing its downstream effectors, are able to bypass its primary activity. As well, mutations in TGFβ receptors (TβRI and TβRII) may lead to the EMT process. On the other hand, loss of TβRI and TβRII is associated with poor prognosis and poor survival.

TGFβ’s signaling can follow either canonical or noncanonical pathways. The canonical signaling pathway proceeds after TGFβ binds with its receptors that activate the SMAD proteins that form a complex in the cell nucleus, acting as a facilitator with DNA-binding cofactors, thus activating the transcription process [58]. On the other hand, the noncanonical pathways utilize several different effectors such as Erk/MAPK, JNK/p38MAPK, and PI3 K/Akt. Interestingly every one of these pathways has a role in EMT [59]. For example, Erk regulates genes involved in cell motility, while overactivation of PI3K leads to cell junction disturbance. Furthermore, loss of TβRI and TβRII are also associated with poor prognosis and poor survival [56] (Figure 6).

#### 2.3.3. Role of CD44

Cancerous cells that undergo EMT present stem-like properties such as the elevated expression of CD44 and their physical transformation in spheres [60]. Epigenetic mutations might be the primer for CSC development [61]. In favor of this argument, we find YAP1 to be an EMT facilitator. YAP1 was found overexpressed in cells that underwent EMT in normal differentiated cells that transitioned into CSCs [62]. This is just one example of many that suggests that the expression of gene-related EMT is a facilitator for CSCs. Furthermore, the TME in which CSCs live pullulates of cytokines and growth factors, favoring their stemness [63]. Furthermore, the hypoxic niche favors the expression of markers such as Nestin. The TME hypoxic state favors the upregulation of Nestin through the activation of the TGFβ-SMAD4 pathway [64].

### 2.4. MicroRNA and CSC

In the past few years, a new and interesting research field has been pursued: the study of microRNAs (miRNA) and their role in PCSC regulation. MicroRNAs seem to play a primary role in the regulation of PCSCs; they seem implicated in regulating the promotion or repression of metastasis. Evidence shows how miR-34a, let-7b, miR-106a, and miR-200 are found in PCa’s progenitor stem cell population [65]. miRNAs are small noncoding RNA molecules that play a role not only in tumorigenesis, but also in cell proliferation, differentiation, and apoptosis [66].

Hypermethylations of miR-34 have been observed in many malignancies with the p53 mutation. Inactivation of both p53 and miR-34a in mouse prostate epithelium causes the PSC compartment, aiding the development of early invasive adenocarcinoma and accelerating EMT-dependent growth, enhancing the self-renewal capability for EMT-dependent growth [67]. In addition, miR-34a is a negative regulator of CD44. CSCs derived from multiple malignant tumors have shown high expression of CD44. Other miRNAs downregulation has been described to have a role in the progression of PCA such as miR-320 and miR-7. The downregulation of the former is associated with increased β-catenin, favoring tumor sphere formation and chemoresistance [37]. Reduced levels of miR-7 correlate with CD133+/CD44+ PCa, and present CSC-like features.

MicroRNAs regulate both CSCs and normal stem cells, but miRNAs dysregulate the process of tumorigenesis. MiR-34a (a p53 target) acts as a key negative regulator of CD44+ in prostate cancer cells and establishes a strong therapeutic agent against prostate CSCs. Furthermore, miR143 and miR-145 suppressed colony formation of PC-3 cells from prostate cancer bone metastasis by inhibiting CSCs properties of PC-3 cancer.

### 2.5. MicroRNA and PCa 

The direct role of microRNA in PCa has been widely investigated.

Walter et al. [68] and. Feng et al. [69] observed downregulation of both miR-148 and miR-152 in high-grade PCa. More specifically, miR-148-3p was found to inhibit PCa cell growth in vitro and in vivo, suggesting its possible role as a tumor suppressor. On the other hand, miR-148b-3p was found to be correlated with PSA and PCA3 in PCa tissue samples, suggesting a potential role as a PCa biomarker [70]. Another interesting study by Ostadrahimi et al. observed a downregulation of miR-185 expression in PCa tissues and cell lines with an upregulation of the antiapoptotic genes *BCL2* and *BCL2L1* [71].

Kristensen et al. evaluated the possible role of miRNAs as diagnostic and prognostic tools in PCa. In their analysis, they developed a prognostic classifier for biochemical recurrence after radical prostatectomy based on mi-R-185-5p, miR-221-3p, and miR-326 [72]. As well, Gurbuz et al. observed increased miR-185-5p expression in patients with elevated PSA levels, especially in patients with PSA > 10 ng/mL. Their study further suggests the potential of miRNAs as prognosis biomarkers.

Overall, microRNA represents an interesting field of research to improve PCa diagnosis, particularly to identify patients at high risk of metastasis and castration-resistant phenotypes [73].

MiRNAs involved with PCSCS and EMT are described in Table 2.

### 2.6. CSC Treatment

PCSC treatments are mainly based on the abovementioned pathways by targeting the microenvironment and immunotherapies.

Specific inhibitors against the Hh pathway (Sonidegib, GANT-61, and GDC-0449), the Wnt pathway (3289-8625, LGK974, Foxy-5, and OMP-54F28), the Notch pathway (RO4929097), and the NFkB pathway (bortezomib, PS1145, BMS345541, Aspyrin, 17-(allylamino)-17-demethoxygeldanamycin, and BKM120) are currently being tested in preclinical and clinical trials [50]. As well, Liu et al. observed that the inhibition of NOTCH1 with shRNA leads to reduction in ABCC1 expression resulting in restored chemosensitivity. Another transporter ABCG2 is also directly involved in the resistance to androgens, and when blocked PCa cells differentiate into androgen-sensitive phenotypes [74].

The PI3K/AKT/mTOR pathway is associated with PCA progression and ADT resistance [75]. Chang and colleagues by the use of the dual PI3K/mTOR inhibitor BEZ235 were able to restore radiosensitivity and induce apoptosis in radioresistant PCSCs [76] PCa.

Some studies suggest the monoclonal antibody Bevacizumab may play a role in modifying TME by reducing tumor neovasculature and disruption of CSC niches. Development of Bevacizumab resistance may be avoided by targeting Rac1 inhibition or P-Rex1 downregulation. The chemosensitivity of PCa cells may also be restored by targeting the CXCR4 receptor, which inhibits sphere formation. On the other side, the development of a castration-resistant phenotype may be triggered by ABC transporters. The idea of targeted differentiation by acting on TME is of great interest and should be confirmed in future experimental studies [77,78]. Different pathways are listed in Table 3. 

## 3. Conclusions

The present review highlights and summarizes the available evidence on prostate cancer and stem cells. CSCs originate from luminal cells with epigenetic modifications mainly driven by the TMPRSS2:ERG complex. CSC development is mainly driven by three signaling pathways: the Wnt, Sonic Hedgehog, and Notch pathways. Overall CSC metabolism is characterized by high levels of oxidated citrate and low levels of Zn. In the development of CSCs, a pivotal role is played by the epithelial mesenchymal transition process which can be activated in a canonical and noncanonical way. As well, several miRNA are involved in this complex process. As previously highlighted, these different pathways and molecules can act as targets for new drugs, opening new strategies in the management and prevention of prostate cancer genesis and progression. CSCs represent a fertile and growing area of research, and ongoing and future studies in the coming years will help the understanding of PCa development and progression.

## Figures and Tables

**Figure 1 ijms-24-07746-f001:**
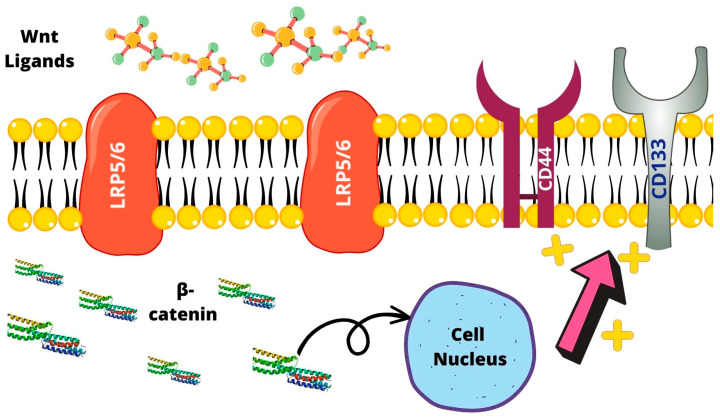
Wnt ligands bind to LRP5/6 receptors accumulating beta-catenin and stimulating its translocation to the nucleus promoting cell survival. This also results in the expression of CD44 and CD133.

**Figure 2 ijms-24-07746-f002:**
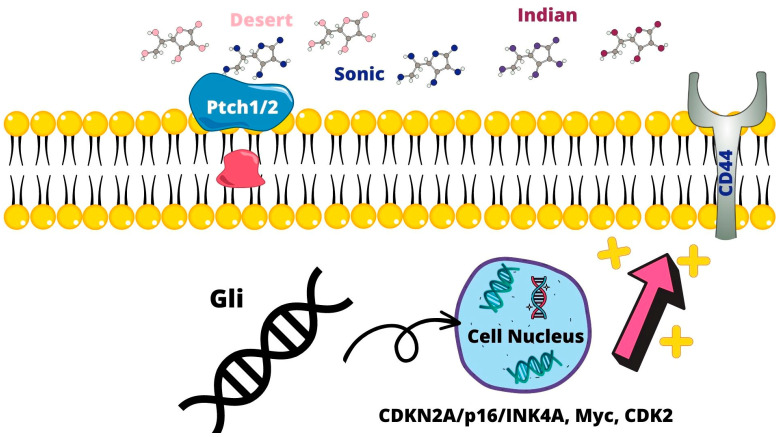
The stimulation of Ptch1/2 receptors by Sonic, Indian, and Desert ligands result in the translocation of Gli into the nucleus, promoting CDKN2A/p16/INK4A, Myc, CDK2, and CD44 expression. This results in an androgen-independent tumor cell growth leading to biochemical and tumor recurrence.

**Figure 3 ijms-24-07746-f003:**
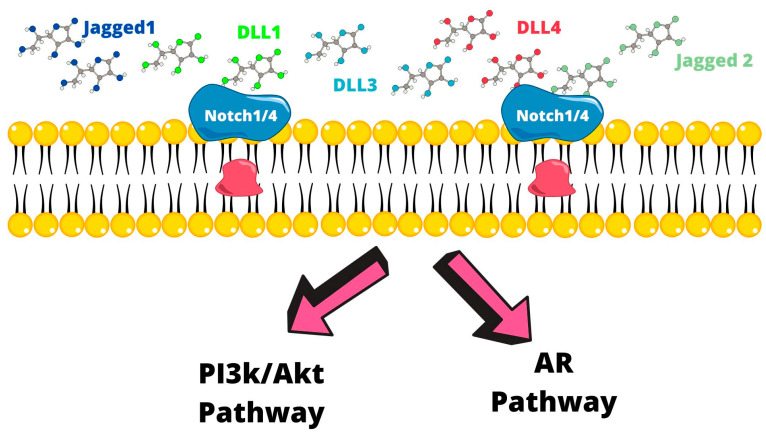
The Notch1/4 receptor is stimulated by DLL1, DLL3, DLL4, Jagged 1, and Jagged 2 ligands, resulting in the activation of the PI3k/AKT pathway and the androgen receptor pathway that promotes survival and progression of PCa cells.

**Figure 4 ijms-24-07746-f004:**
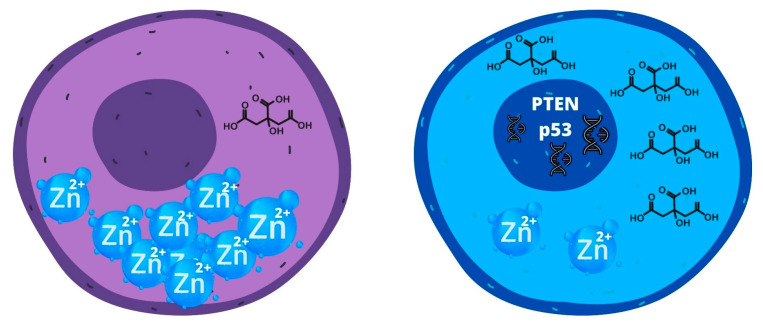
Differences between tumor (blue) and normal cells (purple) are in the Zn^2+^ and citrate concentrations. The high concentrations of citrate and low concentration of Zn^2+^ in tumor cells results in the upregulation of PTEN and p53 genes.

**Figure 5 ijms-24-07746-f005:**
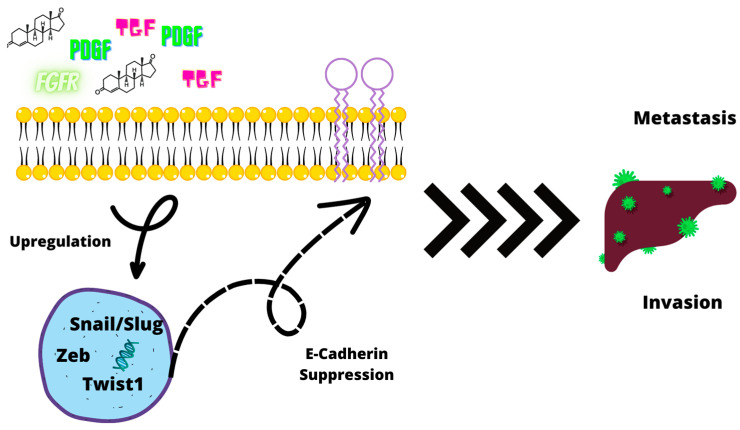
TGF, PDGF, FGFR, and androgens upregulate genes such as Snail/Slug, Zeb, and Twist 1, resulting in E-cadherin suppression. These processes result in an increase in the invasion and metastatic potential of PCa cells through the EMT process.

**Figure 6 ijms-24-07746-f006:**
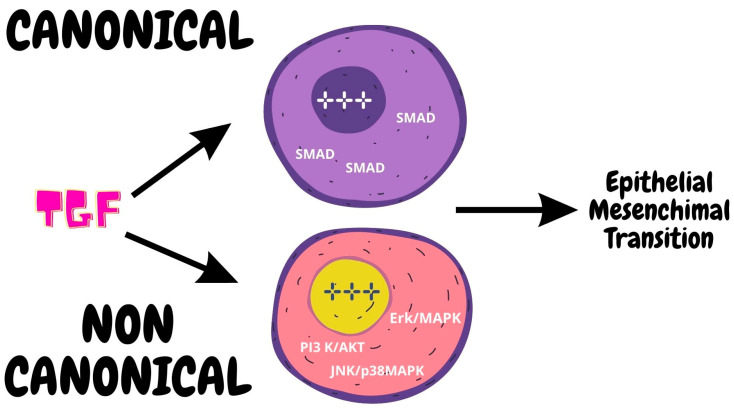
TGF signaling may follow the canonical or the noncanonical pathway. The canonical pathway includes the activation of SMAD molecules activating the transcription process. The noncanonical pathway involves different effectors such as Erk/MAPK, JNK/p38MAPK, and PI3 K/Ak. Both pathways lead to the EMT process.

**Table 1 ijms-24-07746-t001:** Current known PCSC antigens.

Antigen	Function	Role
ABCB1	Transporter	Chemoresistance
ABCg2	Transporter	Chemoresistance, CSC maintenance
ALDH	Aldheyde dehydrogenase	Radioresistance, self-renewal, tumorigenicity
AR-7	Transcriprion factor	EMT, stemness
CD117	Signaling, proliferation, apoptosis, differentiation, migration	CSC maintenance, sphere formation, proliferation, migration, invasion
CD133	Currently unknown	Thepetic resistance, self-renewal, tumorigenicity
CD44	Signaling, Adhesion	Self-renewal, invasion
E-Cadherin	Adhesion, epithelial morphogenesis	Sphere formation
EpCAM	Adhesion, epithelial morphogenesis	CSC maintenance
TG2	Transferase	EMT, chemoresistance

**Table 2 ijms-24-07746-t002:** MiRNAs involved with PCSCS and EMT.

MicroRNA	Function on PCSC	Function on EMT	Targeting EMT-TF	Targeting EMT Pathway	Targeting EMT-CSC
** miR-1 **	/	Inhibition	Slug, Twist1	/	/
**miR-18a**	Progression	/	/	/	/
** miR-21 **	Invasiveness, CRPC	Promotion	/	BTG2, TGFβ	/
** miR-23a-3p **	/	Promotion	E-cadherin	/	/
** miR-23b **	/	Inhibition	/	Src kinase, Akt	/
** miR-25 **	Progression	/	/	/	/
** miR-29b **	/	Inhibition	/	MMP2	/
** miR-30 **	/	Inhibition	/	ERG	/
** miR-32 **	Anti-apoptotic	Promotion	/	BTG2	/
** miR-34a **	/	Inhibition	/	LEF1	/
** miR-34b **	/	Inhibition	/	Akt	/
** miR-100 **	/	Inhibition	/	Aug-02	/
**miR-106**	Progression	/	/	/	/
** miR-124 **	/	Inhibition	Slug	/	/
**miR-125b**	Proliferation	/	/	/	/
**miR-141**	CRPC	/	/	/	/
** miR-145 **	/	Inhibition	ZEB2, HEF1	/	Zeb2
** miR-154 **	/	Inhibition	/	HMGA2, SMAD7	/
** miR-186 **	/	Inhibition	Twist1		/
** miR-195 **	/	Inhibition	/	FGF2, HMGA1, RPS6KB	/
** miR-200 **	/	Inhibition	Zeb1, Zeb2, Slug	/	Notch1
** miR-203 **	/	Inhibition	Zeb2, Bmi1, Survivin CKAP2, LASP1, WASF1, ASAP1 mRNAs	/	/
** miR-205 **	/	Inhibition	Zeb2, Protein Kinase Cε	/	Zeb2, Protein Kinase Cε
** miR-221 **	Proliferation, invasion	/	/	/	/
** miR-222 **	Proliferation, invasion	/	/	/	/
** miR-223 **	/	Inhibition	/	ITGA3, ITGB1	/
** miR-301a **	/	Promotion	/	p63	/
** miR-331-3p **	/	Promotion	/	NRP2, NACC1	/
**miR-375**	Diagnosis	/	/	/	/
** miR-379 **	/	Promotion	/	FOXF2	/
** miR-573 **	/	Inhibition	/	FGFR1	/
**miR-4534**	Tumorigenesis	/	/	/	/

**Table 3 ijms-24-07746-t003:** Drugs and pathways targeting CSCs.

Pathway	Drugs	
Hh pathway	Sonidegib, GANT-61	GDC-0449
Wnt pathway	3289–8625, LGK974,	Foxy-5, OMP-54F28
Notch pathway	RO4929097	shRNA
NFkB pathway	bortezomib, PS1145, BMS345541,	Aspyrin, 17-A-17-DMOD BKM120
PI3K/mTOR	BEZ235	
EMT transformation	Bevaciumab ABC transporter	Anti CXCR4

## Data Availability

No new data were created.

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
