# Peer review of "Cancer Stem Cells and Prostate Cancer: A Narrative Review"

_ijms, 2023, doi:10.3390/ijms24097746_

Round 1

Reviewer 1 Report

This manuscript by Salhi et al., summarizes the information on Prostate cancer and cancer stem cells. Overall, the content seems well-structured and informative. However, there are some issues with grammar and sentence structure that could be improved. Additionally, some sentences are quite long and may be difficult to follow. Here are some specific suggestions:

  • In the introduction, there are a few awkward sentence constructions, such as "Due to population growth and aging..." and "Age, race, genetics, family history, and lifestyle (e.g., obesity, smoking, etc.) represent risk factors..." Consider rephrasing these sentences for clarity.
  • The sentence "Stem cells differ from other cells in both the ability to copiously proliferate (self-renewal) starting from a single cell (clonal) and to differentiate into various kinds of cells and tissue (potent)" is quite long and difficult to follow. Consider breaking it up into smaller, more manageable sentences.
  • In the sentence "These characteristics are very similar to those of cancer cells," it may be clearer to specify which characteristics are being referred to.
  • The sentence "Moreover, PCSCs can also be found in different PCa tissues, such as in metastatic PCa cell lines " could be rephrased for clarity.
  • The sentence "Considering these interesting characteristics of CSCs, their role in PCa development is of great interest to solve the unmet needs in PCa prevention and diagnosis" is difficult to follow. Consider rephrasing it for clarity.
  • In the sentence "An overview of the most important evidence on the topic is essential especially for clinical urologists to have an overview on this complex topic," the use of "overview" twice in the same sentence is a bit repetitive. Consider using a different word in one of the instances.
  • The phrase "PCa possesses an intrinsic heterogeneous nature" could be rephrased for clarity.
  • Line 78 is incomplete
  • While the content in section 2.3 provides some useful information, it is poorly written with a lot of jargon and fragmented sentences, making it hard to understand. Furthermore, the lack of a clear structure makes following the topic's main points challenging.
  • However, it is commendable that the author cites several studies to support their claims.
  • Overall, the content could benefit from better organization and more straightforward language to improve its clarity and readability.

Author Response

Reviewer 1:

This manuscript by Salhi et al., summarizes the information on Prostate cancer and cancer stem cells. Overall, the content seems well-structured and informative. However, there are some issues with grammar and sentence structure that could be improved. Additionally, some sentences are quite long and may be difficult to follow. Here are some specific suggestions:

  • In the introduction, there are a few awkward sentence constructions, such as "Due to population growth and aging..." and "Age, race, genetics, family history, and lifestyle (e.g., obesity, smoking, etc.) represent risk factors..." Consider rephrasing these sentences for clarity.

The sentence has been revised accordingly as follows:

According to the actual statistics, the worldwide prostate cancer burden is expected to grow to almost 2 million new cases and 700 000 deaths in the next 20 years’

  • The sentence "Stem cells differ from other cells in both the ability to copiously proliferate (self-renewal) starting from a single cell (clonal) and to differentiate into various kinds of cells and tissue (potent)" is quite long and difficult to follow. Consider breaking it up into smaller, more manageable sentences.

The sentence has been revised as follows:

Stem cells have the ability to copiously proliferate (self-renewal) starting from a single cell (clonal) and to differentiate into various kinds of cells and tissue (potent)

  • In the sentence "These characteristics are very similar to those of cancer cells," it may be clearer to specify which characteristics are being referred to.

The sentence has been revised as follows:

‘Self-renewal and potency characteristics are very similar to those of cancer cells.’

  • The sentence "Moreover, PCSCs can also be found in different PCa tissues, such as in metastatic PCa cell lines " could be rephrased for clarity.

The sentence has been revised as follows:

‘PCSCs have been isolated in in metastatic PCa cell lines’

  • The sentence "Considering these interesting characteristics of CSCs, their role in PCa development is of great interest to solve the unmet needs in PCa prevention and diagnosis" is difficult to follow. Consider rephrasing it for clarity.

The sentence has been revised as follows:

CSCs may be involved in PCa development and may solve the unmet needs in PCa prevention and diagnosis.

  • In the sentence "An overview of the most important evidence on the topic is essential especially for clinical urologists to have an overview on this complex topic," the use of "overview" twice in the same sentence is a bit repetitive. Consider using a different word in one of the instances.

The sentence has been revised as follows:

‘An overview of the most important evidence on this complex topic is essential especially for clinical urologists.’

  • The phrase "PCa possesses an intrinsic heterogeneous nature" could be rephrased for clarity.

The sentence has been revised as follows:

PCa is an heterogeneous disease where different grades may coexist.

  • Line 78 is incomplete

We appologize for the typo the sentence has been deleted.

  • While the content in section 2.3 provides some useful information, it is poorly written with a lot of jargon and fragmented sentences, making it hard to understand. Furthermore, the lack of a clear structure makes following the topic's main points challenging. However, it is commendable that the author cites several studies to support their claims. Overall, the content could benefit from better organization and more straightforward language to improve its clarity and readability.

We thank the reviewer for his/her comments. The manuscript has been rewritten as suggested specially in section 2.3. We have added some figures for better understanding of the process.

Reviewer 2 Report

The manuscript entitled CANCER STEM CELLS AND PROSTATE CANCER: A NARRATIVE REVIEW by Salhi et al attempts to correlate cancer stem cells and prostate cancer.

The review is not appropriate for publication in the International Journal of Molecular sciences.

The authors attempt to narrate whatever is there in the literature without giving their insights into the field of study. The work does not aim to propagate the important field of study. There is no central theme to the article and looks very disjointed. The paragraphs hardly connect to one another.

They discuss well-known pathways like EMT,microRNAs and the treatment of prostate cancer but do not attempt to unify them as a review, which readers will find interesting.

Reading through the manuscript does not tell many new things that can be utilized by the readers in their studies. There are no diagrams to invoke the interest of the readers of the journal.

I do not recommend its publication in IJMS.

Author Response

The manuscript entitled CANCER STEM CELLS AND PROSTATE CANCER: A NARRATIVE REVIEW by Salhi et al attempts to correlate cancer stem cells and prostate cancer.

The review is not appropriate for publication in the International Journal of Molecular sciences.

The authors attempt to narrate whatever is there in the literature without giving their insights into the field of study. The work does not aim to propagate the important field of study. There is no central theme to the article and looks very disjointed. The paragraphs hardly connect to one another.

They discuss well-known pathways like EMT,microRNAs and the treatment of prostate cancer but do not attempt to unify them as a review, which readers will find interesting.

Reading through the manuscript does not tell many new things that can be utilized by the readers in their studies. There are no diagrams to invoke the interest of the readers of the journal.

I do not recommend its publication in IJMS.

We thank the reviewer for his/her comments. We are sorry the reviewer did not like our review. The review has been improved according to the other reviewers comments improving readability and adding some new figures to help the process. We hope the manuscript now reaches the standards of the IJMS journal.

Reviewer 3 Report

The title of the manuscript is good but the authors should explain about the novility of their work. English language is simple and acceptable. There are some sentences in the main text that should be reformed. Tables are good in quality. Some Figures should be inserted.The multiple and middle-sentence references and sentences without proper reference are the citation problems in this manuscript. Some parts of the manuscript like "Conclusion" have some major problems and should be rewritten.

1. Would you please explain what is the novility of your work?

2. In Page 1, line 2-3

Please reform the title, all words have written with capital letters

3. Page 1, line 14-24

Please rewrite the part "Abstract" totally. It 

can not represent the manuscript briefly and in a good way.

4. All multiple and middle-sentence references in the part "Introduction" should be reconsidered

5. Page 2, line 52-55

Please consider proper reference here

6. Page 2, line 74-75

Why this sentence contains no reference?

7. Page 2, line 90 

Please insert Table 1 right after you mention its name for the first time

8. The part "2.1. Prostate cancer Stem Cells origin" in page 2 and 3 has some multippme references. Please reform all of them

9. The part "2.2. PCSC molecular pathways and metabolism" In page 3 and 4 need one or more simple figure(s) that contains all molecular pathways and metabulism of PCSC. Please insert it (them).

Moreover, all multipple references should be reformed here

10. In Page 4-5, line 184-191

Please insert proper reference in this section.

11. Why line 194-198 in Page 5 has no proper reference(s)?

12. In part " 2.3. Epithelial-Mesenchymal transition and CSC correlation" in Page 4-5. The authors have mentioned some molecular interactions. Please insert some simple figures based on these molecular interactions to make this part more comprehendable.

13. Page 6, line 255-257

Why this part has no reference(s)?

14. Page 6, line 243, part "2.4. Micro-RNA, CSC and PCa"

Please devide this part into two separate part entitled "Micro-RNA and CSC" and " Micro-RNA and PCa" and after that, discuss about them separately.

Furthermore, reform all multipple and middle-sentence references in this part.

15. Page 6 and 7, Table 2

Why this table has no reference(s)??

16. Page 7, part "2.5. CSC and PCa treatment"

Please separate this part into two parts entitled "CSC treatment" and "PCa treatment"

After that, discuss about them separately. Speak about current treatments and future treatments for each one

Besides, reform all multipple references here

17. About the part "Conclusion" in page 8:

Please rewrite this part. Categorize your findings and sort them from the most important one 

18. In page 9, line 359

Why you have inserted references here?

In the part "Conclusion" you should only tell your conclusions based on the previous parts of the manuscript. Please do not insert any references in this part.

19. Please check and adjust the "Reference list" based on the regulations of reference list of journal. (Titles, doi, the name of journal and ... )

Author Response

Reviewer 3

The title of the manuscript is good but the authors should explain about the novility of their work. English language is simple and acceptable. There are some sentences in the main text that should be reformed. Tables are good in quality. Some Figures should be inserted. The multiple and middle-sentence references and sentences without proper reference are the citation problems in this manuscript. Some parts of the manuscript like "Conclusion" have some major problems and should be rewritten.

  1. Would you please explain what is the novility of your work?

We thank the reviewer for his/her comment. The present work is a summary for urologist to have an overview on the topic which is lacking from the literature.

See study aim:

‘An overview of the most important evidence on this complex topic is essential especially for clinical urologists. The aim of the present review is to summarize in a narrative fashion the most relevant data on CSC and PCa.’

  1. In Page 1, line 2-3

Please reform the title, all words have written with capital letters

 We thank the reviewer for his/her comment. Title has been modified accordingly.

  1. Page 1, line 14-24

Please rewrite the part "Abstract" totally. It cannot represent the manuscript briefly and in a good way.

 We thank the reviewer for his/her comment. Abstract has been rewritten.

See new abstract

  1. All multiple and middle-sentence references in the part "Introduction" should be reconsidered

We thank the reviewer for his/her comment. The manuscript has been rewritten accordingly, introduction has been completely reviewed.

  1. Page 2, line 52-55

Please consider proper reference here

We thank the reviewer for his/her comment. References have been updated accordingly.

  1. Page 2, line 74-75

Why this sentence contains no reference?

We thank the reviewer for his/her comment. References have been updated accordingly.

  1. Page 2, line 90 

Please insert Table 1 right after you mention its name for the first time

 Table 1 Has been placed after its mentioned.

  1. The part "2.1. Prostate cancer Stem Cells origin" in page 2 and 3 has some multippme references. Please reform all of them

We thank the reviewer for his/her comment. References have been reformed accordingly.

  1. The part "2.2. PCSC molecular pathways and metabolism" In page 3 and 4 need one or more simple figure(s) that contains all molecular pathways and metabulism of PCSC. Please insert it (them).

Moreover, all multipple references should be reformed here 

We thank the reviewer for his/her comment. References have been reformed accordingly. As well come figures have been added in part 2.2 and 2.3

  1. In Page 4-5, line 184-191

Please insert proper reference in this section.

We thank the reviewer for his/her comment. Reference has been added.

  1. Why line 194-198 in Page 5 has no proper reference(s)?

We thank the reviewer for his/her comment. Reference has been added.

  1. In part " 2.3. Epithelial-Mesenchymal transition and CSC correlation" in Page 4-5. The authors have mentioned some molecular interactions. Please insert some simple figures based on these molecular interactions to make this part more comprehendable.

We thank the reviewer for his/her comment. A figure has been added. See new figures.

  1. Page 6, line 255-257

Why this part has no reference(s)?

We thank the reviewer for his/her comment. Reference has been added.

  1. Page 6, line 243, part "2.4. Micro-RNA, CSC and PCa"

Please devide this part into two separate part entitled "Micro-RNA and CSC" and " Micro-RNA and PCa" and after that, discuss about them separately.

We thank the reviewer for his/her comment.

Furthermore, reform all multipple and middle-sentence references in this part.

  1. Page 6 and 7, Table 2

Why this table has no reference(s)??

Reference has been added

  1. Page 7, part "2.5. CSC and PCa treatment"

Please separate this part into two parts entitled "CSC treatment" and "PCa treatment"

After that, discuss about them separately. Speak about current treatments and future treatments for each one

Besides, reform all multiple references here

We thank the reviewer for his/her comment. We have introduced only CSC treatment in order to stay in the aim of the review. We have added a table for clarity.  References have been reformed accordingly.

  1. About the part "Conclusion" in page 8:

Please rewrite this part. Categorize your findings and sort them from the most important one.

We thank the reviewer for his/her comment. Conclusion has been rewritten.

  1. In page 9, line 359

Why you have inserted references here?

We thank the reviewer for his/her comment. References have been added.

In the part "Conclusion" you should only tell your conclusions based on the previous parts of the manuscript. Please do not insert any references in this part.

We thank the reviewer for his/her comment. Conclusion has been rewritten.

  1. Please check and adjust the "Reference list" based on the regulations of reference list of journal. (Titles, doi, the name of journal and ... )

We thank the reviewer for his/her comment. References have been updated. 

Round 2

Reviewer 2 Report

The diagrams have made the review more confusing. Please remake them and add some more details.

1)None of the figures have figure legends. How would a reader know what the figure wants to convey? Have the authors come across any manuscript in which the figures are not followed by figure legends?

2) Figure 1: what does the blue circle indicate

3) Zinc is a divalent ion. Please show it at Zn2+

4) Figure 5: The invasion metastasis image is not made by the authors. It shows macrophages.

5) Figure 6 is the most ordinary of all. There is an arrow that indicates nowhere. I can see a face in the circle that is underneath the arrow.!!!!!!!!!!!!!!!!

Author Response

The diagrams have made the review more confusing. Please remake them and add some more details.

We thank the reviewer for reviewing our manuscript. Figures have been added as requested and approved by the other two reviewers.

1)None of the figures have figure legends. How would a reader know what the figure wants to convey? Have the authors come across any manuscript in which the figures are not followed by figure legends?

We thank the reviewer for his/her comment and for the possibility to improve our manuscript. Figures are placed just underneath the explanation of the process therefore we did not repeat the process in figure legends to avoid redundance. As per your request these have been specified in figure legends.

See new figure legends.

2) Figure 1: what does the blue circle indicate

We thank the reviewer for the possibility to improve our manuscript. As explained in the text Beta-catenin is translocated in the cell nucleus. The circle represent the nucleus of the cell. It is now better indicated in the figure for clarity.

3) Zinc is a divalent ion. Please show it at Zn2+

We thank the reviewer for the possibility to improve our manuscript. Zinc has been modified as requested.

4) Figure 5: The invasion metastasis image is not made by the authors. It shows macrophages.

We thank the reviewer for his/her comment. We have changed the image as requested, we chose that image to show metastasis potential. The image has been changed as requested and simplified to show the process leading to E-cadherin suppression.  

5) Figure 6 is the most ordinary of all. There is an arrow that indicates nowhere. I can see a face in the circle that is underneath the arrow.!!!!!!!!!!!!!!!!

We thank the reviewer for his/her comment. We have modified to figure to make it clearer. The previous image indicated the cells process of de-differentiation and mesenchymal transition, we apologize if the previous image was not clear enough.

Round 3

Reviewer 2 Report

Accept.